# The Toxic Legacy of Nickel Production and Its Impact on Environmental Health: A Case Study

**DOI:** 10.3390/ijerph21121641

**Published:** 2024-12-10

**Authors:** Jana Levická, Monika Orliková

**Affiliations:** Institute of Social Work and Social Policy, Faculty of Social Sciences, University of St. Cyril and Methodius in Trnava, 917 01 Trnava, Slovakia; jana.levicka@ucm.sk

**Keywords:** nickel, toxic residue landfill, environmental health, environmental burden, profit

## Abstract

Nickel is a chemical element that occurs naturally in soil, water, air, plants, and therefore also in food and other living organisms. However, anthropogenic activities related to the production and processing of nickel can cause its increased concentration in the environment, which is a risk to wildlife and thus to human health. Nickel and its compounds are currently considered immunotoxic and carcinogenic agents that cause a number of health problems. The study examines this problem in the context of the environmental paradigm, which emphasizes the importance of political–economic and socio-economic factors that can seriously affect environmental health. The aim of the study is to draw attention to the economic–political implications of not addressing the environmental burden caused by nickel production and landfilling of waste from this production in Slovakia. The uniqueness of the study is that it reflects the negative impacts of nickel on health in a human–legal context that is characteristic of social work. The study proposes a conceptual model, the verification of which requires further research.

## 1. Introduction

Industrial production has become a self-evident prerequisite for the supply of new goods necessary for human everyday life. Industrial production also comprises activities that provide financial income for the state through taxes. The environmental paradigm requires that industrial activity does not damage the quality of the natural environment, which could have negative consequences for human health. However, economic and political reasons often outweigh the protection of human health. One such case is the processing of nickel ore and the production of nickel products. Nickel as a chemical element can be classified as one of the basic elements that are necessary for several important biological processes such as the healthy growth of plants, animals, and soil microbes [1]. The natural concentration of nickel in water, air or soil is not harmful to humans. However, as a result of anthropogenic activities, nickel concentration in the environment can seriously threaten human health. However, the industrial processing of nickel ore and the production of nickel products can lead to additional exposure to humans and the environment [1,2]. Increased concentrations of nickel in the environment pose a serious risk to human health. Nickel is considered an immunotoxic and carcinogenic agent that can cause health problems such as cardiovascular diseases, asthma, pulmonary fibrosis, respiratory cancer, or contact dermatitis [2,3,4]. Most studies investigating the negative consequences of nickel are carried out by experts in the fields of natural sciences, medicine, or public health. The innovativeness of the study is that it is carried out through the lens of social work, which is anchored in the environmental paradigm. This fact is also the main aim of the study, which is to draw attention to the economic and political implications of not addressing the environmental burden caused by nickel production in Slovakia. The environmental paradigm of social workers obliges them to take an active role in addressing environmental burdens and promoting the right to a healthy environment. Ever-increasing industrial production is also associated with an increase in industrial waste, which can seriously violate this right. The fact that the amount of hazardous and toxic materials also increases with increasing waste [5] q puts pressure on the way it is processed, from landfilling to, for example, burning waste [6], which can also produce toxic substances, resulting in environmental damage. This fact contradicts the long-term mission of social work aimed at achieving an adequate quality of life for as many people as possible [7,8,9] and also with the newer ecosocial paradigm [10,11,12]. The ecosocial paradigm strives to overcome the limited understanding of individually focused case study social work and to move towards a holistic understanding of the interconnectedness of the entire planet with all its species. According to [13], the natural environment is the basis for the fulfillment of many human rights recognized today, which include the right to life, the right to an adequate standard of living, and the right to health. Despite the general agreement on this conceptual framework, the ecosocial paradigm is generally neglected, both at the level of education and research, as well as at the level of direct practice. As stated by [14], the environmental problems that individuals and entire communities are already facing remain outside the framework of social work discourse.


**Theoretical issues**


The natural environment plays a key role not only for all human beings, but also for all living species found on our planet [13]. Man, as a biological species, is existentially dependent on a healthy natural environment, therefore it is not appropriate to separate human problems from environmental problems. The ecosocial approach in social work requires the integration of the natural environment into social work, raising awareness of the risks and injustices caused by environmental problems and contributing to the urgently needed transition to a more sustainable society, e.g., [15,16]. The environmental paradigm is also anchored in the global agenda for social work and social development, which emphasizes two basic principles, which are: environmental justice and environmental sustainability [10]. Social workers should pay adequate attention to all aspects that are somehow connected with the environment. We do not consider it appropriate to narrow the radius of the environmental paradigm only to problems caused by the environmental crisis. We agree with the statement [13], according to which environmental problems refer to a wide range of phenomena, from minor problems associated with climate change to the loss of greenery. The issue of environmental pollution also belongs to this category. Important factors causing environmental pollution include waste, both industrial and municipal. According to the UN, waste is materials that are not primary products (i.e., products produced for the market), for which the producer has no further use in terms of his own purposes of production, transformation, or consumption, and therefore he wants to dispose of them. Waste can be generated during the extraction of raw materials, the processing of raw materials into intermediate products and final products, the consumption of final products, and other human activities. And, of course, waste is also created in ordinary households. According to the directive of the European Parliament, by waste, we mean a movable thing or a substance that its owner disposes of, wants to dispose of, which is in accordance with the Waste Act, or is obliged to dispose of in accordance with special regulations [17]. Pursuant to Act no. 372/2021 Coll. we divide waste into two basic types, namely: hazardous waste and other waste. Ref. [18] further divide waste according to its state into solid, liquid, and gaseous, and according to the place of origin into municipal, industrial, and energy. According to the OECD, hazardous waste is mostly generated during industrial activities and is driven by specific production models. Such waste is of great concern because, if not adequately managed, it poses serious environmental risks: the environmental impact mainly concerns the toxic contamination of soil, water, and air, which seriously damages environmental health [19]. According to [20], the most common method of waste disposal in low- and middle-income countries is landfilling, with most landfills being open, but few being able to be considered sanitary landfills. Despite the fact that landfills seem safe at first glance, they release various impurities into the environment. Landfill biogas, for example, contains approximately 48–56% methane, which contributes to the greenhouse gas effect. Groundwater under or near landfills is also often contaminated with hazardous and toxic waste [20,21,22]. All these products have a negative impact on environmental health. Environmental health is defined as “*the theory and practice of assessing and controlling factors in the environment that may potentially adversely affect the health of present and future generations*” [23]. The original environmental approach to health reflects a predominantly natural science perspective with a focus on the direct, biophysical effects of the environment on human health, i.e., it is oriented towards the protection of human health through regulation and standards. In addition, a critical systems view of environmental health pays attention to the social environment. It recognizes the importance of factors such as clustering, social inequalities, or historical, socio-economic and cultural determinants, while emphasizing the importance of political–economic and socio-economic factors such as deprivation and poverty and psychosocial processes that influence health [24].

## 2. Materials and Methods

In the article, we present part of the results obtained from a more broadly conceived piece of research focused on environmental justice in the context of social work. The Slovak Republic (hereafter SR) is literally littered with a large number of environmental burdens that threaten the health and life of its inhabitants. In Figure 1, created by the State Geological Institute of Dionýz Štúr (hereafter referred to as ŠGÚDŠ), it is possible to see the coverage of SR by environmental loads. The causes of their occurrence are different, but their common element is the long-term failure to solve problems associated with negative impacts on environmental health.

The aim of the study is to draw attention to the economic and political context of not addressing the environmental burden caused by nickel production in Slovakia. Understanding the economic and political context will help to obtain a better understanding of the situation surrounding the leaching residue landfill, which has been known for more than 60 years to have a negative impact on environmental health, and yet which has not yet been disposed of.

The research questions to which we were looking for an answer were: How and why was Nickel Smelter Sereď created? How is it possible that, despite the knowledge about the harmfulness of nickel to the environment, production continued for so long? Why, even 30 years after the end of production, was the leaching residue waste dump not disposed of?

From the possible research designs, the authors decided to use a case study [25,26], which allows an intensive systematic study of a chosen subject, in the context of a given situation, while researchers respect the social environment, culture, and interactions that arise in the system in which the subject under investigation is a part. Thus, the case study makes it possible to grasp, explore, describe, and explain the phenomenon that is the focus of the researcher’s attention [27]. Several authors state that the case study does not have an established methodology that would precisely determine the sequence of specific steps or sub-procedures. This fact provides the researcher with a high degree of flexibility [28], but at the same time it places demands on him or her to define as precisely as possible the goal, context, space, etc., in which the case study is carried out [29,30,31,32,33,34,35,36].

In our case, the analyzed unit [37] was the leaching residue landfill, which is located on the outskirts of the city of Sereď. The case study was realized in historical and political context [32]. The case study was spatially and temporally bounded [33,34,35,36]. Spatially, the study was limited to the wider surroundings of the city of Sereď, in which the landfill is located. This is an area approximately 50 km from the landfill. The research was limited in time by the years 1962–2024. The reason for such a wide time range is the start of production in the nickel smelter in Sered, which took place in 1962 and the ongoing impacts on the quality of the environment and environmental health in the region today. The upper time interface is given by the year 2024, i.e., the current time.

The main methods for data collection were a content analysis of secondary data, a self-constructed questionnaire, and semi-structured interviews. The subject of the content analysis were the reports of institutions that carried out research on the landfill or in its surroundings, professional and scientific journals, books, official websites of the Ministry of the Environment of the Slovak Republic (hereafter referred to as the Ministry of the Environment of the Slovak Republic), websites of the city of Sered, local newspapers, and NGO websites, which have been striving for years to achieve the elimination of the leachate landfill. The interviews, which lasted between 30 and 45 min, were transcribed, and qualitative thematic analysis was used to process them [38,39]. The research sample consisted of 12 informants, of which 9 were men and 3 were women. This research set was created using the snow-ball technique. The self-designed questionnaire was published online. The data were collected in the period March–April 2024. The research group consisted of 354 respondents, of which 274 (77.40%) were women and 80 men (22.60%).

The data obtained by the questionnaire are presented through descriptive statistics. The obtained data were processed to jointly create a text that provides answers to the research questions.

## 3. Results

We have obtained the information necessary for plotting the history of the leaching residue landfill through a content analysis of secondary data. The analyzed materials were reports of institutions that carried out research on the landfill or in its vicinity, official websites of the Ministry of the Environment, local newspapers, and websites of NGOs that have been striving for years to achieve the elimination of the leaching residue landfill.

### 3.1. The Reasons for the Establishment of Nickel Smelter Sered

The Nickel Smelter Sered was established in 1962 as a state enterprise. At that time, Slovakia, as part of the Czechoslovak Socialist Republic, was a member of the Council for Mutual Economic Assistance (Comecon). Comecon was a multilateral international organization based in Moscow. When it was created, there were originally six Eastern European countries in it. However, their number grew rapidly. The main goal of Comecon was the coordination of the economic development, scientific, and research development of the member states, as well as the coordination of their investment and trade policy, which was based on the principles of central planning [40,41,42]. The decision to start nickel production in Sered was the result of Comecon’s decision, which knew that Czechoslovakia could handle the production of nickel, with which it would supply its member states.

### 3.2. Factors Sustaining Nickel Production in Sered

The benefits of starting nickel production can be traced both domestically and internationally. Nickel-bearing ore was imported from Albania, which at the time was a member of Comecon. According to preserved documents, Albanian ore contained only less than 1% of nickel, and the remaining 99% consisted of other metals and impurities. In addition to nickel, electrolytic cobalt was also produced from Albanian ore. The entire production system required demanding and harmful chemical processes. Currently, up to 50% of iron is in the waste from the Albanian ore known as lúženec (leaching residue) [43,44]. It is possible to assume that without the coordination of Comecon, Albania would not be able to sell this ore. Another fact that needs to be taken into account is Comecon’s effort to establish itself in international trade. Any export of member countries’ products outside the Comecon states corresponded to this effort [45,46,47]. Czechoslovakia sold nickel to all Comecon countries at relatively low prices. It also made a profit from the sale of nickel to the countries of Western Europe, where it sold nickel at significantly higher prices. Despite this, the nickel smelter (probably due to the guaranteed prices for the Comecon countries) failed to make a profit, and the state subsidized it annually from CZK 180 million to 250 million [43].

In the early years, not all the negative effects of nickel production were known. This is probably why a wastewater treatment plant was not built in the nickel smelter, and the company did not even have its own sewage system, but was connected to the municipal sewage system [48].

The start of production at the nickel smelter in Sered brought 1500 jobs, which enabled the residents of Sered to obtain work close to where they live. The nickel smelter offered its employees housing under very advantageous conditions, which was a benefit especially for young people. “*I was young, I got married two years before and we were expecting our first child. There was a shortage of flats, and Niklovka (Nickel smelter) provided company flats, which were practically free…*” (i3). Although it was later confirmed that nickel has not only negative genotoxic, but also embryotoxic, carcinogenic, and teratogenic effects on the body [3,43,48], the state authorities managed to hide this information for a long time. Nevertheless, the government ordered that, due to the negative impact of nickel on human health, employees had to undergo an examination by the company doctor every six months. The company built its own health center where, in addition to a general practitioner, a dentist, dental technician, and gynecologist worked. The center had biochemical laboratories and a health service with ambulances that worked non-stop. Extensive health care was perceived by the employees as an advantage. “*I went to Niklovka to work also because it had good health care. You know, back then every enterprise had its own doctor. You did not have to pay for the examination, as before the war. And we had a doctor right in the enterprise…*” (i2). The availability of various health professionals is highlighted by its former employees even today. “*No, I don’t know about any negatives of Niklovka. It has built flats, a kindergarten here, the employees had a doctor for free, directly in Niklovka…. Today, we have to go to doctors who knows where and everything is paid for, prescriptions, specialists…*” (i1). The company tried to compensate for the possible negative impact on the health of employees with social benefits. “*Niklovka provided its employees with above-standard benefits. It had several recreation centers and I have gone to the sea with my family almost for free a few times... what others would have paid for it at that time…*” (i5). The fact is that the nickel smelter significantly contributed to building the infrastructure of the city of Sereď. Near the workplace, the company built an entire housing estate, crèches, two kindergartens, a stadium, a cultural center, and it even built several publicly accessible wells. Niklovka had its own school, which trained locksmiths, electricians, operating electrical engineers, and mechanics of measuring devices for the needs of the company. The company also operated a secondary industrial school for workers, where the employees of the company during the so-called evening studies could complete their secondary education in the areas needed by the company [45]. The company Niklová huta was perceived very positively in Sered. It contributed to the development of the city in various ways, and the management of the company did not hide information about the riskiness of the workplace. “*We knew that there was some risk, it was even stated in the recruitment leaflets that we would receive a risk bonus. It wasn’t much, but everyone welcomed the extra money*” (i12). The question is what information the management had. The nickel smelter’s investments in the city’s infrastructure and the care of its employees also affect the perception of one of its remnants—the leaching residuals landfill, which is waste from the processing of Albanian ore.

### 3.3. Barriers to Liquidation of the Luženec Landfill

Among the barriers that prevent a faster pace of liquidation of the landfill, one can include insufficient legislation, the lack of competences of the city of Sereď, the economic interests of the landfill owner and the attitude of the public living in its vicinity. Public indifference, often caused by ignorance, helps maintain the status quo. The findings of the questionnaire research point to a low level of awareness among respondents, which was also reflected in the respondents’ approach to other aspects of solving the situation.

Through questionnaire research (Figure 2) we found that only 89 respondents (25.14%) consider the waste in the landfill to be toxic, and as many as 61 (17.23%) respondents think that the waste in the landfill is safe.

One of the reasons why the respondents do not consider it necessary to follow the information about the leaching residue landfill near Sered may be the fact that the Ministry of the Interior of the Slovak Republic has reclassified the site, which was originally used as a tailings pond, into a landfill, which gives the public a sense of security. “*Well, I don’t think that the landfill is dangerous today… When production was running, maybe then. Even the ministry is no longer talking about a dust dump, but rather a landfill…*” (i8). The impression of the safety of the landfill is also strengthened by its partial reclamation. “*Look how beautifully green it is. If it were to be harmful, nothing would grow there and not trees and bushes…*” (i6).

The state stopped financing the production of nickel in Sered in 1991, i.e., two years after the change in economic and political conditions in Czechoslovakia. The production lasted until the processing of the last stocks of Albanian ore, which happened in 1992. A year later, the property of the Nickel smelter in Sered was sold to private hands. In 1993, the Nickel smelter in Sered was privatized. Since there was no interest in buying it as a whole, the company was divided into several segments. One of these segments was also a landfill, which was created from waste from the production of nickel. Figure 3 captures the gradual change of landfill owners.

At the same time, 126 (35.59%) women and 50 (14.12%) men consider their own information about the landfill to be good, which is a total of 176 (49.72%) respondents. As many as 43 (12.15%) respondents have no interest in information regarding the leaching residuals landfill (Figure 4). 

The nickel smelter, as a state-owned enterprise, was not only a producer of nickel and electrolytic cobalt, but it was also an institution responsible for the creation and management of a waste dump of by-products from the processing of ore imported from Albania. The company produced up to **300,000 tons** of fine metal dust annually [43,48]. During the sale of the landfill, which was created as a by-product of ore processing, the new owner’s obligation in relation to environmental protection was included in the sales contract. With each additional sale, this commitment decreased until it disappeared completely. A huge number of chemicals were used in the production of nickel, such as ammonia, sodium sulfide, sodium sulfate, hydrochloric acid, and others. Already in 1994, the district office in Galant issued a regulation that obliged the first owner, ABH Hell, to recultivate the landfill, which was to be implemented by 1999. After the change of the owner, it issued another regulation, according to which the landfill was to be recultivated by 2002. From a legal point of view, these decisions bound the specific owner who owned the landfill at that time. The new owners did not invest any funds in the reclamation of the landfill, arguing that the regulation only applied to the original owner. Finally, in 2004–2005, partial reclamation works were carried out by the city of Sered at its own expense. The city reforested the slopes of the landfill and built a drainage system on the landfill. Currently, the landfill contains about **6.5 million tons** of waste in the form of fine dust. Opinions about the amount of stored lye vary; some authors state that there are up to **8.5 million tons** of lye waste at the landfill [48,49,50].

Individual governments, also thanks to ecological activists, were informed about the problems caused by ecological burdens. Since 2006, the Ministry of the Environment, in cooperation with the State Geological Institute of Dionýz Štúr (hereafter referred to as ŠGÚDŠ), the Slovak Environmental Agency (hereafter referred to as SAŽP), and other organizations, have been systematically monitoring the state of individual locations where such a burden is located. ŠGÚDŠ also carried out a survey at the leaching residuals landfill near Sered. On the basis of the conducted survey, it was established that the landfill is located in a protected water management area, which belongs to the protection zone of water resources, the protection zone of natural healing resources, and sources of natural mineral waters. The landfill is located in the basin of the *Malý Dunaj*, *Váh, Dunaj*, and *Derňa rivers*. As stated in the report, the landfill is located over an area with significant underground water reserves. There is no natural protection in the site and the wider site of the landfill. The bottom of the landfill is uninsulated, unreinforced, as a result of which the threat to groundwater is very high. Surveys carried out at the initiative of the city of Sered or citizens’ initiatives confirmed that the waste stored at the landfill contains high concentrations of nickel, chromium, and cobalt, which far exceed the intervention criteria. High concentrations of nickel and chlorine were also confirmed in the surrounding soil and in the waters in close proximity to the landfill, where a high concentration of NH4+ was also confirmed, and the water also contains petroleum substances [48,51,52]. The survey confirmed that soil contamination is so high that it poses a significant risk and limits further land use. Despite this finding, agricultural land was not removed from the register of agricultural land [48].

Contrary to these findings, the information on the official website of the Ministry of the Interior of the Slovak Republic states that the occurrence of these substances does not exceed the permitted limits. Monitoring of the nickel smelter’s impact on the environment and human health began only in the 1990s. The negative impact of nickel production cannot be considered a thing of the past. Polymetallic dust from the landfill still pollutes the environment around it. According to the conducted studies, this dust spreads up to a distance of 50 km. Annually, the wind blows away about 600 tons of fine dust from the landfill. There are no studies on the effects of this landfill on the surroundings organized and realized by the state [53]. Monitoring as part of their research was performed, for example, by ŠGÚDŠ, or SAŽP, which, however, focused only on the landfill. Partial data from the monitoring, which are available in the Register of Environmental Loads in the Slovak Republic, indicate that a deeper monitoring of the landfill will begin in November 2013 and end in June 2020 [54]. This information was entered into the Register in 2012. *The data from the implemented monitoring are not public.*

As [55] writes, no one concealed from the employees that it was a risky work environment. Employees were informed that they must wear respirators, and in some workplaces they wore leather aprons, glasses, and special masks in the lye. However, it is true that some employees ignored these regulations, worked without respirators, or smoked in the workplace. However, they considered only the interior of the company to be a risky environment. As one of our respondents said, “*I never thought that the fields around Niklovka could be harmful…several of us had a garden there. The director also had a garden a short distance from ours. If that factory caused such problems as they say today, we probably wouldn’t have those gardens there…*” (i4). “*We had a garden there for years. If you saw those carrots, or peppers… whatever you planted, when you cared, you had a nice harvest. I think it’s just talk, we’re all healthy, even the grandchildren…*” (i10).

It is therefore not surprising that, as part of quantitatively oriented research, it was found that only 180 (50.85%) respondents know that landfills can threaten their health, despite the civil initiative From Black to Green, which fights for the elimination of landfills.

We were interested (Figure 5), in what opinion the respondents have about the removal of the landfill and what argument they would give for its removal. It is surprising that 55 (15.54%) respondents would say that the reason for removing the landfill is that it spoils the appearance of the landscape. All respondents in this case were women. We can assume that their opinion was influenced by their residence, from which they have a view of the leaching residuals landfill. Despite the seriousness of the impact of the landfill on environmental health, up to 29 (8.19%) respondents think that the landfill does not need to be removed.

We consider it interesting to note that respondents aged 18–25 consider the landfill to be only a remnant of socialism. Only respondents aged 36 and older associated the landfill with a negative impact on health.

The effort to eliminate the leaching residuals landfill has been present in the city of Sered for a long time. The municipal office itself carries out various activities within its competences, and in some of them it also joins with NGOs, which also strive to achieve change. Thanks to these initiatives, current residents of the region have access to sufficient information about the negative effects of the landfill on environmental health. We therefore investigated whether the respondents are aware of the danger associated with the leachate landfill (Figure 6).

As many as 92 (25.99%) respondents could not assess whether the landfill poses any danger to them. A total of 156 (44.07%) respondents considered the landfill to be toxic or very harmful, which is not even half of the respondents. 

As many as 68 (19.21%) respondents answered negatively to our question about whether the respondents are interested in the further fate of the landfill. A total of 98 (20.68%) respondents answered that they are only partially interested in the landfill. Interest in the further fate of the landfill was confirmed by 178 (50.28%) respondents (Figure 7).

This relatively indifferent approach to the environmental condition of the environment in which the respondents live is also confirmed by the fact that only 56 (15.82%) of them participated in some activity aimed at improving their environment (Figure 8).

The indifference of some respondents may also be caused by the greater distance of their residence from the landfill as they do not realize that the alkaline dust affects areas more than 50 km from the landfill site. The pressure to change the situation thus remains in the hands of environmental activists coming mainly directly from Sered.

The city of Sered already came up with a landfill liquidation project in 2002. It was an ambitious project that envisaged the disposal of the landfill within 10 years. The partner of the project was to be the German company WAX, which guaranteed to transport 600,000 tons of slag from Sered annually. The owner of the landfill had the opportunity to participate in the project as the main contractor, but he refused after a promising start.

The state lost its influence on the removal of this environmental burden by agreeing to the privatization of its originator, the Sereď Nickel Smelter, without the necessary legislative guarantee.

The further fate of the landfill is in the hands of its current owner. Based on our findings, we developed a conceptual model [56] (Figure 9), which can also be applied in the search for solutions to other old environmental burdens in Slovakia.

## 4. Discussion

Waste, which in the past was considered unusable, is now considered an economic commodity thanks to new knowledge and technologies. Nevertheless, Ref. [57] state that about 70% of municipal solid waste ends up in various landfills.

Ref. [58] also draw attention to the gradual increase in electronic and electrical waste. This type of waste is an important source of recycling, as it contains metals such as copper, aluminum, iron, or steel. They are all precious metals, the reserves of which are exhaustible in nature. Ref. [59] also draw attention to the other side of electronic and electrical waste, according to which this waste also contains heavy metals such as lead, cadmium, and arsenic, which are harmful to health. However, some light metals, such as nickel, also pose a health risk, which at elevated concentrations have been confirmed to act as an immunotoxic and carcinogenic agent [2,3,4].

In the context of social responsibility, it is necessary to solve the question of how to process waste, which is, on the one hand, a source of necessary substances and, on the other hand, threatens environmental health through its processing. From the perspective of social work, the authors state that work in landfills and the handling of waste stored in landfills can be perceived as the result of unfairly distributed power.

Those who cause environmental pollution benefit from the production that produces the waste. However, waste producers are minimally involved in the removal and processing of this, often dangerous and even toxic, waste. The claim that organized waste disposal creates job opportunities underlines the social injustice that manifests itself in the fact that people with low education, who are also socially excluded, are involved in waste disposal activities [59] and they perform their work for minimal financial income. Lack of education is the cause of poor waste management. This fact can be considered a global problem, the consequence of which is environmental contamination and strengthening of social exclusion [60]. In our case, common residents of the Sered region work on the liquidation of the leaching residue landfill, and their salaries correspond to usual salaries for workers and drivers in this area. Since their health status is not monitored in any way, it cannot be ruled out that their health is threatened not only by the content of materials contained in the landfill waste, but also by rodents, flies, and other insects that live in landfills and that can transmit various infectious diseases, as described by [61]. In addition, the landfill near Sered has a negative impact on the health of people who live within a radius of approximately 50 km from the landfill, which represents approximately 1 million people.

Ref. [9] states that the destruction of the environment is the result of the pursuit of economic wealth by a part of society, which reinforces economic injustice. This statement is in line with our findings, as we found that the main reason for the slow liquidation of the leaching residue landfill near Sered is the reluctance of the current owner. The Ferroport Company withdrew from the project developed by the city of Sereď, because this solution would bring it a lower profit than when they sell the leaching residuals on its own [62]. It should be noted that this landfill is not currently a worthless pile of waste. The leaching residue contains 46–53% iron and several other usable elements such as silicon, chromium, and magnesium. If we realize that the estimated amount of the leaching residue in the landfill is between 6.5 and 8.5 million tons, it is a relatively valuable property. As the owner of Ferroport stated, in 2004 they sold 1000 tons of the leaching residue and wanted to sell another 4000 tons by the end of the year. Experts claim that at this rate, the liquidation of the leaching residue landfill will take approximately 600 years [62]. This is a huge period of time during which the health of countless residents of the region may be at risk.

The toxicity of the leaching residue landfill has already been confirmed, which was the reason for its cultivation by the city of Sereď. The gradual slow sale of leaching residue waste also means the “opening” of a partially rehabilitated landfill. Ferroport is gradually liquidating the forested upper part, which facilitates easier penetration of alkaline waste into the environment.

Our case study confirms the findings of [63] that the environmental crisis needs to be reflected in the context of social justice, which belongs to the domains of social work.

### 4.1. Strengths and Limitations 

This study is unique in Slovakia as similar research, focused on a systemic approach to the so-called old environmental burdens, i.e., to burdens whose foundations are in socialism, has not been realized yet. The environmental paradigm itself is not sufficiently known in the Slovak environment. The case study was focused only on one location, which may mean that some facts are tied only to the location where the landfill is located. However, we consider the conceptual model created by us to be usable in solving other environmental burdens. The study unequivocally confirms a strong role of the state in the creation of environmental burdens, but also its obligations in their removal.

### 4.2. Ethical Aspects of the Study

During the implementation of the case study, the ethical principles of scientific work were strictly followed, from truthful information to research participants, voluntary participation of respondents, anonymity of respondents, and objectivity in data processing. This study was financially supported by the project: APVV-20-0094 Environmental justice in the context of social work.

## 5. Conclusions

The operation in Nickel Smelter Sered was finished in 1992, but to this day it pollutes the environment in the wide outskirts of the city of Sered. The reasons for this state of affairs must also be sought in the deep past, as the government of Czechoslovakia assumed the disposal of waste from the processing of Albanian ore. Memorialists claim that research was also carried out by the company, which also investigated the possibilities of further processing of the leaching residuals. Some experiments were even carried out. However, at that time, the use of waste proved to be too financially demanding and not very effective. The whole pile of leaching residue is rightly called a toxic legacy of socialism [49].

Another reason was the lack of legislation that would have sufficiently protected the environmental health of the Slovak population at the time of privatization after 1990. In Slovakia, we have more than 2000 registered loads and thousands of illegal landfills, which threaten the environmental health of not only the residents of the Slovak Republic, but also the residents of neighboring countries.

In our research, we found out that the main cause of the slow liquidation of the leaching residuals landfill near Sered can be attributed to the reluctance of the current owner to participate in the project with which the city of Sered wanted to speed up the liquidation of the landfill.

The failure to solve the problems caused by environmental burdens in the Slovak Republic is also helped by the fact that individual scientific fields investigate this issue only in the context of their scientific field. Ecological, environmental, and natural sciences are dominant in this area. Social work brings into the discourse on environmental burdens the discourse on issues of the right to a healthy environment or environmental justice. The potential of the social work profession lies precisely in its possible contribution to opening discourse in an interdisciplinary context and “bringing” environmental problems into public discourse.

## Figures and Tables

**Figure 1 ijerph-21-01641-f001:**
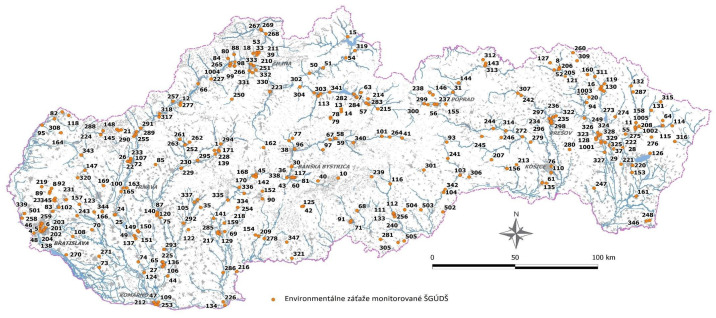
Map of environmental burdens in Slovakia, Source: ŠGÚDŠ, 2021.

**Figure 2 ijerph-21-01641-f002:**
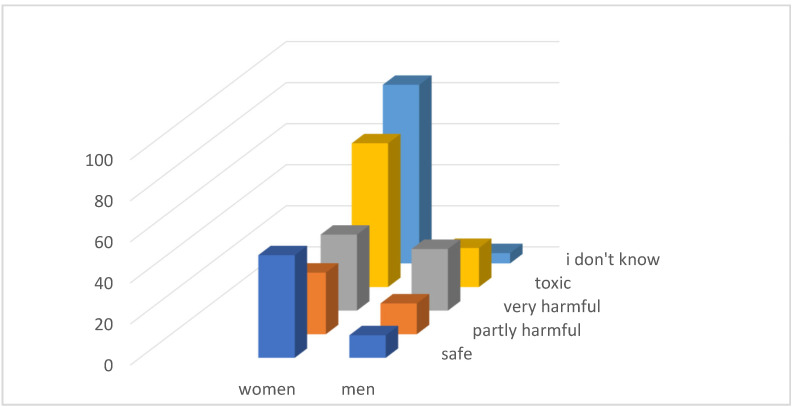
Evaluation of the safety of the leaching residue landfill.

**Figure 3 ijerph-21-01641-f003:**
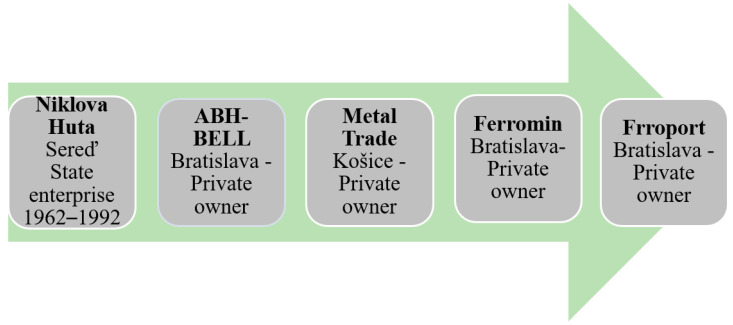
Overview of the change of owners of the leaching residue landfill near Sered.

**Figure 4 ijerph-21-01641-f004:**
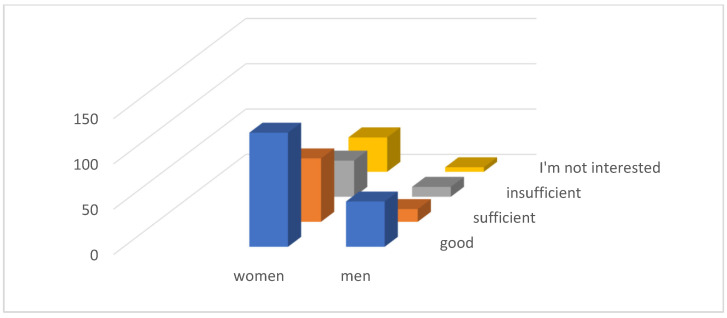
Information of respondents.

**Figure 5 ijerph-21-01641-f005:**
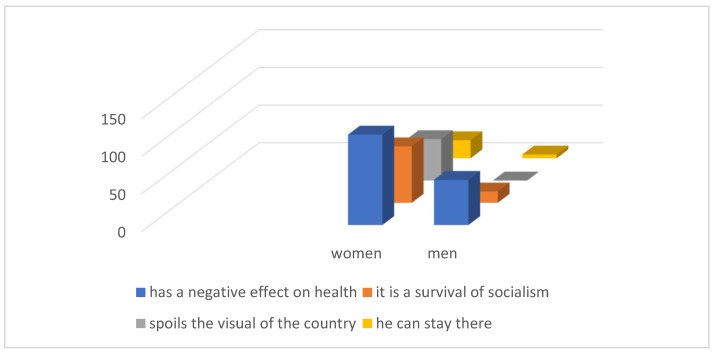
Opinions on landfill removal.

**Figure 6 ijerph-21-01641-f006:**
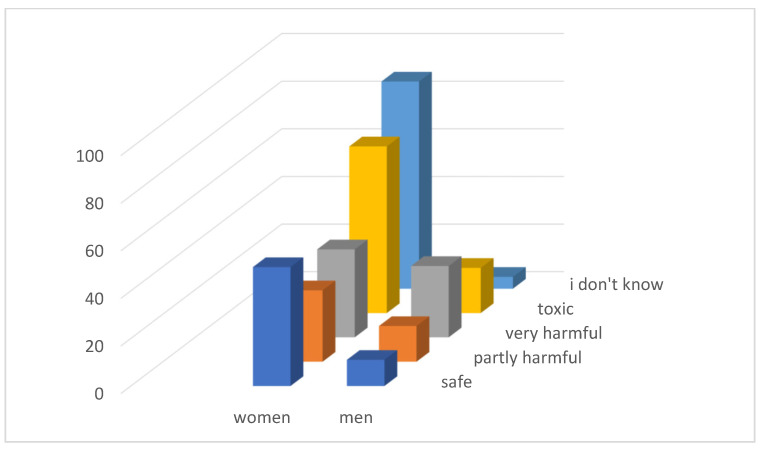
Perception of the danger associated with the landfill.

**Figure 7 ijerph-21-01641-f007:**
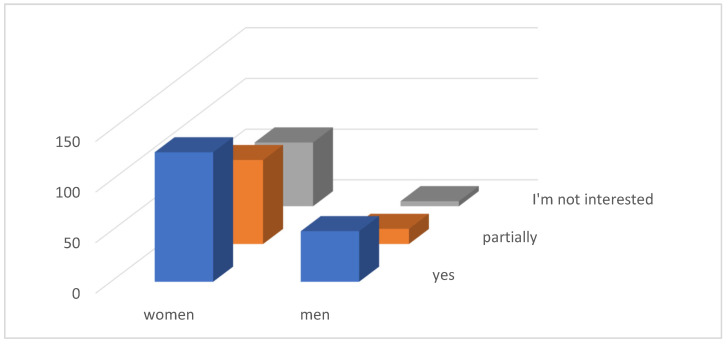
Interest in the further fate of the landfill.

**Figure 8 ijerph-21-01641-f008:**
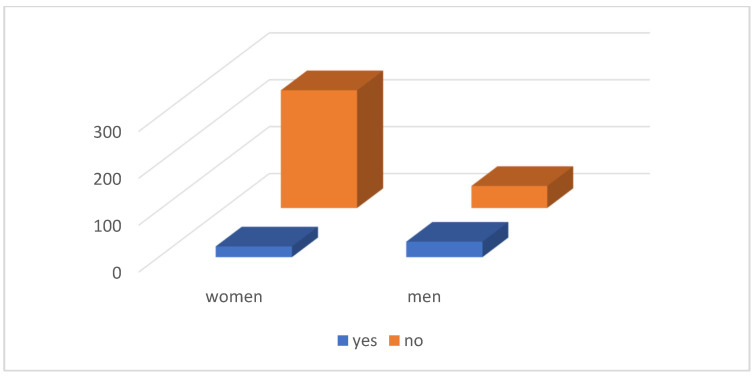
Participation of respondents in activities focused on the environment.

**Figure 9 ijerph-21-01641-f009:**
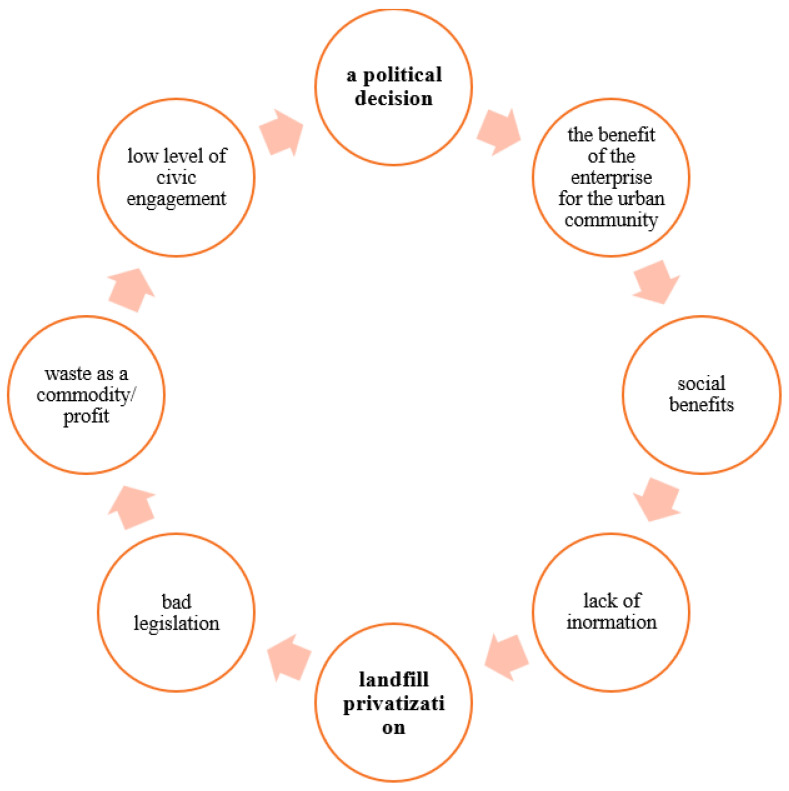
Conceptual model.

## Data Availability

Inquiries regarding the availability of the data included in this study may be directed to the corresponding author.

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
