# Peer review of "The Toxic Legacy of Nickel Production and Its Impact on Environmental Health: A Case Study"

_ijerph, 2024, doi:10.3390/ijerph21121641_

Round 1
Reviewer 1 Report
Comments and Suggestions for Authors
This study develops a conceptual model through a case study that analyzes the toxic legacy of nickel production and its impact on environmental health. The investigation focused on the formation process of toxic waste landfills and their persistence in the environment. However, The structure of the Abstract and Introduction in the manuscript is not very well. I suggest to be rejected or a major revision after careful presenting the novelity of the article. Several concerns require careful attention:
- The methods of data collection and the quantitative aspects of this case study are overstated in the Abstract section; the conceptual model needs to be introduced.
- The logic in the Introduction section lacks clarity and does not align with the title; additional information on the environmental impact of the nickel industry should be included.
- The innovative aspects of the presented work should be emphasized in the Introduction.
- The application of the conceptual model should be described.
- The Results section is overly verbose and should be simplified; unnecessary dialogue references (e.g., lines 355-360) should be removed.
- Graph 1 should be renumbered as Figure 2, with subsequent figures renumbered accordingly.
- The style of the figures should be diversified (e.g., incorporating pie charts, line charts, etc.).
- Detailed descriptions should be provided in the comments for each diagram.
- The legend in Graph 6 should be relocated to the axis.
- The text in Figure 3 should be placed within the boundaries of the image.
- The formatting of references should be standardized (e.g., references 19, 25, etc.).
Quality of English Language need to be further improved.
Author Response
Pozrite si prosím prílohu.Comments 1: The methods of data collection and the quantitative aspects of this case study are
overstated in the Abstract section; the conceptual model needs to be introduced.
Response 1: At the reviewer's suggestion, we revised the abstract. The study as a whole was
not carried out in a conceptual model, we only developed a draft of a conceptual model at the
end, as recommended by Naeem, Ozuem, 2022, the validity of which needs to be verified by
further research.
Comments 2: The logic in the Introduction section lacks clarity and does not align with the
title; additional information on the environmental impact of the nickel industry should be
included.
Response 2: We have also revised the Introduction, so as to emphasize more the negative
consequences of nickel on the environment and human health p. 1-2
Comments 3: The innovative aspects of the presented work should be emphasized in the
Introduction.
Odpoveď 3: We thank the reviewer for the comment to emphasize the innovative aspects of the
study in the abstract, we have incorporated it. p. 1 -2
Komentár 4: The application of the conceptual model should be described
Odpoveď 4: Thanks for the warning. The study as a whole was not carried out in a conceptual
model, we only developed a draft of a conceptual model at the end, as recommended by Naeem,
Ozuem, 2022, the validity of which needs to be verified by further research..
Comments 5: . The Results section is overly verbose and should be simplified; unnecessary
dialogue references (e.g., lines 355-360) should be removed.
Response: The reviewer suggests reducing or possibly eliminating the findings from semistructured interviews. However, these represent the opinions and attitudes of residents, which
we have included among the obstacles to solving the landfill. We are afraid that without them
this fact would not be visible
Comments 6: Graph 1 should be renumbered as Figure 2, with subsequent figures renumbered
accordingly.
Response 6: We accepted the request to change the name graph -figure. p. 8, 9, 10, 12,13,14,15
Comments 7: . The style of the figures should be diversified (e.g., incorporating pie charts, line
charts, etc.).
Response 7: We believe that the style of graphs does not affect the quality of our research
Comments 8: Detailed descriptions should be provided in the comments for each diagram.
Response 8: We agree that the description of the individual graphs/figures seems relatively
brief, nevertheless, we think that the descriptions are clear enough
Comments 9: The legend in Graph 6 should be relocated to the axis.
Response 9: Thank you for pointing out that the legend under graph 6 is off-axis. We have
renamed the graph to figure 8 and removed the error on page 14
Comments 10: The text in Figure 3 should be placed within the boundaries of the image.
Response 10: We thank you for the kind warning that the text in Figure 3 is out of scope - we
have corrected it. Figure 3 has been renamed to Figure 9. p. 15
Comments 11: We are thanks the reviewer for the comment. The formatting of references
should be standardized (e.g., references 19, 25, etc.).
Response 11: At the reviewer's suggestion, we checked the formatting of the links p. 19
References.
We thank the reviewer for the time he devoted to our study and the development of suggestions
for improving its quality

Reviewer 2 Report
Comments and Suggestions for Authors
It was a pleasure reading your work and about your specific case of study. Is very sad that people are not aware enough and engaged enough to solve such issue.
My general comments are in the line of improving the graphs. They could look better using other software to produce them.
Also, I suggest you make a more streamlined discussion of your findings. I think you should organize it in clreare subsections and summarize your findings ina more precise way.
Author Response
Comments 1:My general comments are in the line of improving the graphs. They could look better using other software to produce them.
Response 1: We have formally edited the text in this section to make it clear to which research question the presented results relate. We modified figures no. 4 p. 10 and 9. p. 15
Comments 2: Also, I suggest you make a more streamlined discussion of your findings. I think you should organize it in clreare subsections and summarize your findings ina more precise way.
Response 2: We have formally edited the text in this section to make it clear which research question the presented results relate to. p. 6 and 8
We thank the reviewer for the time he/she devoted to our study and developing suggestions to improve its quality.
Reviewer 3 Report
Comments and Suggestions for Authors
Dear Author,
-The subject of the article is an important issue that needs to be focused on. However, there are serious problems with the way the manuscript is organized. First of all, the introduction should be reconsidered and unnecessary topics should be removed. The subject is about nickel pollution and the harms of nickel, but this is not mentioned in the introduction. A section about nickel's environmental pollution and harms to human health should definitely be added in the introduction. It should be especially stated that it passes from soil to food in high amounts and accumulates in food, and that nickel is a serious threat to human health. I will also recommend an article for this. This article states both the harmful effects of nickel and that it passes from soil to food in high amounts, and (in the conclusion) that it is one of the biggest problems for human health in the future.
This article contains a lot of information about the harms of nickel and the amounts in foods.
Suggested manuscript: Elemental analysis and health risk assessment of different hazelnut varieties (Corylus avellana L.) collected from Giresun – Turkey. https://doi.org/10.1016/j.jfca.2023.105475.
-In addition, the aim and importance of the study should be clearly written in the introduction. The aim written between lines 130-132 should be moved to the introduction and the importance of the study should be stated.
-The language of the study should be a little more academic. Instead of expressions such as "we" or "our study" used in many places, expressions such as "this study" should be used. The writing style should be reviewed in general.
-In the conclusion section:
"The reasons for the establishment of Nickel Smelter Sered"
-The title and lines 181-191 can be moved to the introduction section because they are not relevant to the conclusion. These can be written instead of unnecessary information in the introduction section.
-The conclusion section should only be about presenting the results obtained in an orderly manner. Theoretical information should be in the previous sections.
-The article I suggested can also be used in the sentence in lines 220-221 below. Because this article mentions the carcinogenic and non-carcinogenic toxic effects of nickel:
Although it was later confirmed that nickel has not only negative genotoxic, but also embryotoxic, carcinogenic and teratogenic effects on the body [39, 44]
-Please add references to places where information is given, such as line 297.
"The company produced up to 300,000 tons of fine metal dust annually." (Ref?)
---In addition---
-For abbreviations such as "GDP" used in the abstract, write the full name before the abbreviation at the first use and then write the abbreviation. Then only the abbreviation can be used.
-Line 51; move the comma to the correct place.
-Line 58; should be corrected.
"with all its species [9] (Stamm, 2023). According to [9], ...."
(Stamm, 2023) in the first sentence should be removed and the second sentence should start with According to (Stamm, 2023).
Author Response
Comments 1, 2:
The subject of the article is an important issue that needs to be focused on. However, there are serious problems with the way the manuscript is organized. First of all, the introduction should be reconsidered and unnecessary topics should be removed. The subject is about nickel pollution and the harms of nickel, but this is not mentioned in the introduction. A section about nickel's environmental pollution and harms to human health should definitely be added in the introduction. It should be especially stated that it passes from soil to food in high amounts and accumulates in food, and that nickel is a serious threat to human health. I will also recommend an article for this. This article states both the harmful effects of nickel and that it passes from soil to food in high amounts, and (in the conclusion) that it is one of the biggest problems for human health in the future.
Response 1: At the reviewer's suggestion, we have revised the Introduction to emphasize the negative impacts of nickel on the environment and human health. p. 2-3
Response 2: Thank you for pointing out the source ŞEKER, M. E. 2023, we have read the study with interest and incorporated it into our text p. 3, 8, 15
Comments 3: In addition, the aim and importance of the study should be clearly written in the introduction. The aim written between lines 130-132 should be moved to the introduction and the importance of the study should be stated.
Response 3: We have also accepted the reviewer's recommendation to present the aim of the study more clearly in the Introduction. p. 1-2
Comments 4: The language of the study should be a little more academic. Instead of expressions such as "we" or "our study" used in many places, expressions such as "this study" should be used. The writing style should be reviewed in general.
Response: 4: We have also accepted the recommendation regarding the language. p. 5,6,7
Comments 5: We have added the missing reference/source. Please add references to places where information is given, such as line 297. "The company produced up to 300,000 tons of fine metal dust annually." (Ref?)
Response: We have added the missing reference/source. p.10, line 314
Comments 6: When editing the abstract, we have removed the abbreviation HDP from it.
Response: When editing the abstract, we have removed the abbreviation HDP from it.p.1
Comments 7: Line 58; should be corrected
Response: We have corrected formal errors in lines 58 p.3(changed lines 100)
Comments 8: The title and lines 181-191 can be moved to the introduction section because they are not relevant to the conclusion. These can be written instead of unnecessary information in the introduction section.
Response 8: We have decided to keep the text on lines 181 to 191 (changed lines 185-196) because this text is related to the first stage of our research.
We thank the reviewer for the time he/she devoted to our study and developing suggestions for improving its quality
Round 2
Reviewer 2 Report
Comments and Suggestions for Authors
Dear authors,
It was a pleasure reading your revised version.
My only comment, is that since all your figures are original from your own work, please do not include the "Source: Own 2024" it is nto needed at all.
Great job!
Author Response
Comments: My only comment, is that since all your figures are original from your own work, please do not include the "Source: Own 2024" it is nto needed at all.
Response: We have accepted your comments and removed the sources for the figures.

Reviewer 3 Report
Comments and Suggestions for Authors
Dear Authors,
The requested corrections have been made.
Kind Regards
Author Response
Thank you for your opinion and review.